# NO_2_ Sensing Behavior of Compacted Chemically Treated Multi-Walled Carbon Nanotubes

**DOI:** 10.3390/mi13091495

**Published:** 2022-09-08

**Authors:** Nikita I. Lapekin, Valeriy V. Golovakhin, Ekaterina Yu. Kim, Alexander G. Bannov

**Affiliations:** Department of Chemistry and Chemical Engineering, Novosibirsk State Technical University, 630073 Novosibirsk, Russia

**Keywords:** gas sensors, sensor, carbon nanotubes, response, sensitivity, NO_2_, functionalization, chemical treatment, multi-walled carbon nanotubes

## Abstract

This article is devoted to the investigation of the sensing behavior of chemically treated multi-walled carbon nanotubes (MWNTs) at room temperature. Chemical treatment of MWNTs was carried out with a solution of either sulfuric or chromic acids. The materials obtained were investigated by transmission electron microscopy, scanning electron microscopy, Raman-spectroscopy, X-ray diffraction, and Fourier transform infrared spectroscopy. The active layer of chemiresistive gas sensors was obtained by cold pressing (compaction) at 11 MPa of powders of bare and treated multi-walled carbon nanotubes. The sensing properties of pellets were investigated using a custom dynamic type of station at room temperature (25 ± 2 °C). Detection of NO_2_ was performed in synthetic air (79 vol% N_2_, 21 vol% O_2_). It was found that the chemical treatment significantly affects the sensing properties of multi-walled carbon nanotubes, which is indicated by increasing the response of the sensors toward 100–500 ppm NO_2_ and lower concentrations.

## 1. Introduction

The development of effective techniques with rapid, sensitive, and selective detection capabilities for hazardous and toxic gases in industry and everyday life is of interest from the point of view of environmental protection and occupational safety. This is because exposure to such gases, even at low concentrations, can cause visual impairment, respiratory illness, and even death [1,2,3,4].

There are several ways of detecting gases in the air. One of the most commonly used methods of analysis is gas chromatography-mass spectrometry because of its reliability and accuracy [5,6]. However, this method has a number of disadvantages, such as high cost and cumbersome equipment, as well as the complexity of the process of pre-concentration of samples [7,8,9], which prevents monitoring and analysis of the gas content in the air in real time. Due to the problems described above, this analysis method is not suitable for detecting dangerous gases in real time. Chemiresistive gas sensors are preferable for detecting gases due to the obvious advantages, including low cost, high sensitivity, and the possibility of embedding in portable devices. Furthermore, with the development of MEMS, chemical sensors can be miniaturized and integrated into intelligent applications, in particular smartphones, smart watches, and hand-held medical instruments [10,11]. Thus, it is possible to provide reliable monitoring of the content of dangerous gases in real time.

Traditional semiconductor sensors consume a lot of power for operating at high temperatures. As a rule, the gas sensors are implemented in the form of semiconductor sensors based on metal oxides, but they operate at relatively high temperatures (above 200–350 °C) [12]. In view of this, the creation of sensors operating at room temperature is an urgent task that requires a solution coming soon.

A wide range of nanomaterials, including carbon nanomaterials, metals, metal oxides or sulfides, and organic semiconductors, offers opportunities for the development of novel gas sensors. Recently, much attention has been paid in this field of carbon nanomaterials (CNMs), which have found their wide application in catalysis, as well as active materials for biosensors [13,14,15,16,17] and chemiresistive gas sensors [18,19,20,21,22]. There are a lot of carbon nanomaterials used for the detection of NO_2_, e.g., carbon nanotubes (namely multi-walled carbon nanotubes (MWNTs) and single-walled carbon nanotubes (SWNTs)) [23,24,25,26,27,28], carbon nanofibers [29], graphene [30,31], and their hybrids [32]. These materials could be used independently or in combination with other non-carbon materials: SiO_2_/MWNTs [26], ZnO/SWNTs [33], SnO_2_/MWNTs, Sm_2_O_3_@ZrO_2_/CNTs/GCE ([34]) etc. This is due to the unique physical structure and excellent electronic properties. MWNTs are a special type of material that can be used to detect dangerous gases [35,36,37]. Carbon nanomaterials can be implemented in sensors either as films [38,39] or as compacts [40]. The method of preparation greatly affects the properties of the sensors.

The preparation of films is a more time-consuming technique. This is due to the need to disperse the solution of the material by ultrasound, as well as subsequent filtration and drying [19]. It should also be noted that the measurement of electrical properties, in this case, is difficult since it is impossible to obtain data on the precise geometric parameters of the films. Another problem is the reproducibility of obtaining films when films with different properties are obtained using the same laboratory technique from the same material (for example, spin coating and drop casting). Therefore, the use of compacted samples is preferable in order to enhance the sensing properties of carbon nanotubes.

One of the ways to improve the sensitivity of these materials is chemical treatment (functionalization). There are several methods of functionalization. Plasma oxidation is a rather fast process, lasting from milliseconds to several seconds [41]. It is carried out at low and atmospheric pressure, and at room temperature. One of the advantages of plasma treatment is that the morphology of nanomaterial is particularly preserved [42]. In [43], treatment of MWNTs in inductively coupled plasma produces mainly carbonyl and ester groups. However, this method is rather energy intensive and has a low degree of functionalization (C:O). Gas-phase oxidation is the oxidation of MWNTs by water vapor at relatively low temperatures (up to 400 °C). In [44], it was shown that this method works only for the CVD synthesis of MWNTs. In this case, the structure has many defects, and the oxygen content was 42 wt% in the form of various functional groups.

Additionally, the chemical treatment makes it possible to purify carbon nanotubes. Concentrated and dilute mineral acids [45], mixtures of various acids, and mixtures of acids with chromium or manganese salts [46] are used as oxidizing agents. After such treatment, hydroxyl, carboxyl, and lactone groups are most often formed on the surface of nanotubes. The disadvantage of this method is the duration of the process (up to 24 h or more) and low yield of treatment. However, the degree of functionalization is much higher than that of plasma treatment usually.

The functionalization of carbon nanotubes (CNTs) was already used for the creation of NO_2_ gas sensors, but it was based on the creation of films [25,47]. This work aimed at the preparation of compacted MWNTs, which were chemically treated, and it is devoted to the investigation of the sensing behavior of chemically treated multi-walled carbon nanotubes at room temperature (25 ± 2 °C) towards NO_2_.

## 2. Materials and Methods

### 2.1. Non-Treated Materials

Multi-walled carbon nanotubes were chosen as a starting material to study the effect of chemical treatment on the gas-sensitive properties of chemiresistive sensors. MWNTs of 1020 and 4060 grades were obtained by Shenzhen Nano-Tech Port Co. (Shenzhen, China). Detailed data on the synthesis method of MWNT-1020 and MWNT-4060 samples are not available since these materials are commercial products.

### 2.2. Chemical Treatment

The surface of the MWNT samples was modified using a chemical treatment (Figure 1). The following acids were used as oxidizing agents (the concentration is mentioned in brackets): H_2_SO_4_ (6 M), H_2_Cr_2_O_7_ (6 M), and H_2_Cr_2_O_7_ (3 M). A 6 M sulfuric acid solution was prepared from concentrated sulfuric acid of 94.6% purity. H_2_Cr_2_O_7_ solutions were prepared from solid chemically pure chromium oxide (6) of 99.0% purity.

A sample of untreated MWNTs weighing 0.15 g was placed in a conical flask (250 mL), poured with 100 mL of acid, and heated at 80 °C with constant stirring for 6 h. Then, the obtained samples were washed with distilled water and filtered using a vacuum filtration unit. After filtration, the samples were placed in a furnace for 12 h at 100 °C. Then, the samples were ground in a mortar and sifted through a sieve with a mesh size of 100 microns.

In the process of oxidation, various kinds of functional groups [45,48] can be formed (Figure 2):

### 2.3. Preparation of Sensing Material

The active layer of the gas sensor was obtained by pressing untreated and modified MWNT samples. Usually, the active materials of chemiresistive gas sensors are the films [3,49], but we use a completely different approach in order to show that CNTs can also be used for gas sensing after compaction. Pressing of powders was carried out in a circular mold at room temperature (cold pressing, 25 ± 2 °C). The pressure was 11 MPa, and compaction was carried out for 30 min under constant pressure (Figure 3). The mold consisted of a base, two matrices, a shell, and a punch. The procedure of tableting consisted of selecting fractions <0.08 mm in size, placing the shell with the lower matrix on the lower base, filling the material and its even distribution throughout the shell, placing the upper matrix and punch, and direct pressing. Compaction was carried out without a binder. Bare and chemically treated MWNTs made it possible to form a mechanically stable tablet.

The tablet obtained was used for nitrogen dioxide detection at room temperature. Two contacts (silver paste) were deposited on the surface of the table. The two-point technique was used to test the sensor response.

### 2.4. Characterization of MWNTs

Changes caused by metamorphosis in the structure of MWNTs, including chemical modification, were investigated using the following methods.

#### 2.4.1. Transmission (TEM) and Scanning Electron Microscopy (SEM)

Transmission electron microscopy was carried out using JEM-2010 (Jeol, Tokyo, Japan). The elemental composition of powder samples treated with MWNTs was studied using a S-3400N scanning electron microscope (Hitachi, Tokyo, Japan). The add-on for energy dispersive X-ray spectroscopy manufactured by “Oxford Instruments” was used in order to analyze the composition of elements in carbon materials and the degree of functionalization. A study of the elemental composition of the sample’s surface was carried out without sputtering (energy of electron beam 10 keV), Li-Si detector at an elevation of detector at 35 degrees, and inclination of sample 0°.

#### 2.4.2. Raman Spectroscopy

The structural features of defectiveness of carbon nanomaterials were determined by Raman spectroscopy on a T64000 (Horiba Jobin Yvon, France) (λ = 514 nm). The degree of disorderliness of the carbon materials (degree of defectiveness) was estimated from the ratio of the intensities of D and G peaks.

#### 2.4.3. X-ray Diffraction (XRD)

In addition to Raman spectroscopy, the structural features of the carbon nanomaterials were also determined by X-ray diffraction. Based on the value of the FWHM, which corresponds to the interlayer spacing between the graphene layers, the degrees of graphitization (*y*) of the carbon nanomaterials were calculated by the formula [50].
*y* = ((3.44 − *d*_002_)/(3.44 − 3.354)) × 100%,(1)
where *d*_002_ is the interlayer spacing, nm.

#### 2.4.4. Fourier Transform Infrared Spectroscopy (FTIR) and X-ray Photoelectron Spectroscopy (XPS)

Determination of functional groups in treated MWNTs was carried out using an FTIR spectrometer FT-801 (Simex, Novosibirsk, Russia) (TU-4434-805-59962935-2019) with a reflection attachment for studying powdered samples PO-45N with an angle of incidence of 45°, with a bottom sample location and built-in visualization system.

The technique for studying initial MWNTs consisted of taking a baseline of the infrared neutral substance, e.g., KBr, and mixing it with the sample to make it less absorbing.

X-ray photoelectron spectroscopy was carried out using X-ray photoelectron spectrometer (SPECS Surface Nano Analysis GmbH, Germany). Spectrometer was equipped with the PHOIBOS-150 hemispherical analyzer and the source of X-ray radiation XR-50M with Al/Ag anode. Al Kα (hv = 1486.74 eV) monochromized radiation was used.

### 2.5. Sensing Measurement

The gas sensing properties of the obtained sensors were investigated on a specialized gas installation (Figure 4). The installation includes gas lines for the analyzed gases: nitrogen dioxide (5000 ppm NO_2_ in the air) and carrier gas, synthetic air (79 vol% N_2_, 21 vol% O_2_). 

The main sensor parameter was the sensor response (%):ΔR/R_0_ = (R − R_0_)/R_0_ × 100%,(2)
where R—sensor resistance when exposed to sample gas, Ω; R_0_—sensor resistance when exposed to synthetic air, Ω.

The flow rate of the gas mixture supplied to the measuring cell for contact with the sensors was 100 mL/min. The concentration of the analyzed gas in the system was regulated by the flow mass controllers of the air-analytical mixture coming from the cylinders. The analyte flow was changed so that the resulting mixture had a certain gas concentration (10–1000 ppm).

The measurements were performed using the following technique: before starting the experiment, a closed, empty cell was purged with carrier gas for 10 min at a flow rate of 100 mL/min. Then the sample was placed in the cell and degassed at a heating temperature of 70 °C and a carrier gas flow rate of 100 mL/min for 10 min. These steps were designed to clean the gas delivery system and the measuring cell, desorb the compounds from the sensor surface and reduce the relative humidity inside the cell to 18%. The next step was to measure the baseline, while the carrier gas was fed for 60 min at a flow rate of 100 mL/min. Then, the analyte (NO_2_) of a certain concentration was fed for 10 min into the cell, and then pure carrier gas was fed for 10 min to purge the system and recover the sensor. The scheme of measuring cell is presented in Figure 5.

Studies of the sensor response to NO_2_ exposure were performed at room temperature (25 ± 2 °C) in the concentration range of 100–500 ppm.

## 3. Results and Discussion

### 3.1. Characterization of MWNTs

MWNT-1020 is a combination of fairly thick carbon nanotubes and chain-like carbon nanofibers 60–80 nm in diameter, with a narrow hollow channel 10–20 nm in size (Figure 6). The absence of catalyst confirmed that the sample was chemically treated by manufacturer. SEM images showed that the carbon nanotubes were strongly aggregated (Figure 6b,d).

MWNT-4060 sample was represented by long nanotubes 40–100 nm in diameter, with many walls and a hollow channel of small diameter (Figure 7). The inclusions of nanoparticles of catalyst showed that the sample was not chemically treated by manufacturer compared to the MWNT-1020 sample.

Two typical peaks are in the Raman spectra of the carbon samples studied, i.e., D peak (~1350 cm^−1^) corresponding to the disordered structure of the carbon material and G peak (~1590–1610 cm^−1^) corresponding to the ordered structure of carbon in the sp^2^ hybridized state [52] (Figure 8).

The ratio of peak intensities characterizes the disorder degree of the samples under study [53,54]. The smaller the peak ratio, the lower the degree of imperfection of MWNTs. All the data on Raman spectroscopy are presented in Table 1. For all types of treated samples, the defectiveness of MWNT-1020 was reduced to ~25% after treatment. The defectiveness of the MWNT-4060 sample, on the contrary, increases up to 2.6 times compared to MWNT-1020. This difference in chemical treatment is caused the removal of surface graphene layers from the MWNT-1020 sample. This sample was already chemically treated by the manufacturer; therefore, the additional treatment induced the strong “etching” of surface layers. At the same time, the non-treated MWNT-4060 sample supplied by the manufacturer were oxidized strongly, which in turn induced the growth of I(D)/I(G) compared to untreated ones.

Figure 9 shows the X-ray diffraction patterns of the studied samples. The patterns showed the same curves, an indication of the presence of reflections related to carbon material without any impurities. Strong 002 plane reflection could give us a possibility to calculate the graphitization degree of the materials.

Table 2 shows the data of treatment of X-ray diffraction patterns values of the interlayer spacing d_002_, as well as the degree of graphitization of the studied carbon nanomaterials.

Based on the data presented in the table, it can be concluded that the structure of the MWCNT-1020 sample is more disordered, which, among other things, correlates with the Raman spectroscopy data.

Table 3 shows the results of energy-dispersive X-ray spectroscopy (EDX) and XPS.

The lower the C:O ratio, the better the functionalization of the sample. This is caused by the formation of C–O, C=O, and C(O)O groups, the presence of which is usual for functionalized CNTs [12]. The elements contained in the acids used were observed. It is assumed that the acid anions could adsorb in the pores of the MWNTs or form new covalent bonds at the defect site. We also observe the presence of chromium in the samples treated with dichromic acid; this may be related to the adsorption of the dichromate ion, hence there was such high oxygen content, together with the formed functional groups.

The difference between the results obtained using EDX and XPS is different due to the fact that EDX analyzes the depth of the material, and XPS is used to investigate only the surface (up to 8 nm in depth). From the XPS, one can say that the treatment in sulfuric acid for MWNT-4060 produced many sulfur-containing groups, such as sulfonic acid groups (-SO_3_H), on the surface.

Based on the obtained FTIR spectra in Figure 10, one can make the following conclusion: the absorption band at 685–762 cm^−1^ corresponds to the presence of C-H bonds; for the original samples, the presence of these vibrations is absent.

The 1132–1405 cm^−1^ absorption bands corresponded to vibrations of C-O bonds of different natures (alcohols, esters, etc.). However, such vibrations are observed only for samples MWNT-4060, most likely because they are more defective, and the formation of C-O bonds to the carbon atoms passes easier to come to a more favorable stereochemical position. The absorption band at 1634 cm^−1^ corresponded to vibrations of C=O bonds, but the vibrations can also correspond to C=C bonds.

### 3.2. Gas Sensing Behavior

Response curves of chemically treated and compacted MWNTs are shown in Figure 11. It was found that chemical treatment in acids led to an increase in the response value for the samples MWNT-1020 and MWNT-4060 at 100–500 ppm NO_2_. The negative slope of the curves is associated with a decrease in the resistance of the materials during NO_2_ feeding. This is related to the fact that the adsorption of the electron acceptor molecule NO_2_ on the surface of MWNTs induces the charge carrier transfer from the nanotube, which increases the concentration of the latter and increases the conductivity.

A short description of ΔR/R_0_ of compacted MWNT samples is given in Table 4. Chemical functionalization leads to the formation of additional defects and an increase in the number of charge carriers, which is reflected in an increase in conductivity and response.

For MWNT-1020, treatment in sulfuric acid had the best effect, which is reflected in an increase in the response of 100–500 ppm NO_2_ from 1.7–13.0% to 5.2–15.1%. The sensor recovery at the time of carrier gas fed after 250 and 500 ppm NO_2_ was also observed, as evidenced by the increase in the response curve. The response time of the sensors is shown in Table 5.

Treatment in dichromic acids also positively affects the sensor properties of MWNTs. Interestingly, treatment in more concentrated dichromic acid (6 M) is less effective compared to treatment in less concentrated acids (3 M). The response of nanotubes treated in H_2_Cr_2_O_7_ (3 M) was higher than that of nanotubes treated in H_2_Cr_2_O_7_ (6 M). When the properties of MWNTs treated in dichromic acids are compared with those of the original MWNTs, it can be concluded that the response of the treated MWNTs increases in a range between 100 and 250 ppm, and the response at 500 ppm is less for the MWNT samples treated in dichromic acids.

For MWNT-4060, treatment in acids resulted in a slight increase in the response threshold at 100–500 ppm NO_2_. The highest response was observed for the sample treated in concentrated dichromic acid. This is caused by the low degree of functionalization of the material. Since the MWNT-1020 sample was preliminarily oxidized by the manufacturer, it possessed a lower C:O ratio and therefore showed a better response compared to initially non-treated MWNT-4060 sample. This fact is also confirmed by the resistance of tablets obtained after the compaction of MWNT powders. This value was higher for MWNT-1020 samples compared to MWNT-4060. The DC resistance at 25 °C ranged from 1.39 to 5.14 Ω. Resistance of compacted MWNTs (R_25 °C_) increased in all cases after chemical treatment of MWNTs, excluding the MWNT-4060 sample treated with H_2_SO_4_ (6 M). 

If we compare the different marks of MWNTs, one can note that chemical treatment in similar acids has different effects on the sensing properties. This is due to the different nature of the starting materials. For example, the MWNT-1020 sample was initially etched with acids, which is confirmed by Raman spectroscopy data, namely, the degree of disorder. For the MWNT-1020 sample, the value of the peak ratio I(D)/I(G) is lower than for MWNT-4060—1.04 and 0.56, respectively. These values are complemented by the graphitization degree obtained by XRD. The graphitization degrees of MWNT-1020 and MWNT-4060 were 46.5% and 61.6%, respectively. Subsequent treatment of MWNT-1020 in acids led to a decrease in the disorder degree, which is explained by the attachment of functional groups to the defects. It is also worth noting that in the case of treatment in dichromic acids, adsorption of dichromate ions is observed, which is confirmed by the high chromium content according to EDX data. Adsorption of dichromate ions leads to a decreased response to NO_2_ due to a decrease in free centers for adsorption of the NO_2_ molecule. 

The highest response was observed for the MWNT-1020 sample treated with sulfuric acid. Treatment of MWNT-4060 in 3 M dichromic acid led to the formation of more defects, which is confirmed by the abnormally high value of the I(D)/I(G) ratio, 1.46. This fact is also confirmed by EDX. Treatment in sulfuric acid also leads to the formation of defects, but adsorption of sulfate ions was not observed. The best treatment in terms of sensing properties for MWNT-4060 is treatment in concentrated dichromic acid.

The cross-sensitivity (selectivity estimation) of sensors was examined (Appendix A*).* These sensors were tested for the detection of model gases (ammonia and methane) at room temperature. The selection of ammonia was based on electron-donating interaction with MWNTs (that is different compared to the NO_2_ sensing mechanism). From the graphs, one can see that these sensors had a low response to 100–500 ppm of ammonia compared to 100–500 ppm of nitrogen dioxide (Appendix A). It is also worth noting that the plots of change in sensor resistance at the time of ammonia feeding have a positive slope, while the plots of change in sensor resistance at the time of nitrogen dioxide show a negative slope. This is explained by the fact that the adsorption of ammonia on the surface of the MWNTs reduces the number of charge carriers, which contributes to an increase in the resistance of material. Chemical modification of the initial MWNT samples led to a change in the sensing properties. Chemically modified MWNT-1020 showed a greater response compared to the non-treated one. Treatment in 6 M sulfuric acid and 3 M dichromic acid increased the response approximately by a factor of 3. This is due to the fact that treatment in these acids increases the number of functional groups, which is reflected in an increase in the degree of graphitization and an increase in the C to O ratio (according to EDX and XPS). Chemical modification of MWNT-4060 in sulfuric and dichromic acids (6 M) also led to an improvement in the sensor response.

A comparison of methane detection by compacts showed that there was no significant response to CH_4_ at room temperature at 100–500 ppm concentration (Appendix A). The absence of response was also detected for chemically treated samples.

The active materials studied in this work for the detection of NO_2_ can be used in the chemical and petrochemical industry in case of accidents or failures for the detection of nitrogen dioxide at room temperature. The possibility of creating a sensor using only a tablet consisting of MWNTs is an advantage compared to films since the techniques of film preparation (e.g., drop casting, spin coating, etc.) are not appropriate for industrial applications. At the same time, cold isostatic pressing is a simple technique used in this work, which makes it possible to obtain the reproducible sensing behavior without using a binder for compaction of MWNT powders.

## 4. Conclusions

Compacted materials based on chemically treated MWNTs’ powders were obtained. It was suggested that the changes in gas sensing properties based on multi-walled carbon nanotubes are caused by the following mechanisms:(1)Generation of functional groups that act as sorption centers.(2)Formation of new defects.(3)Adsorption of ions on the surface of MWNTs.

The chemical modification resulted in a change in the elemental composition (mainly the increase in oxygen concentration), which influenced the improvement of the gas sensing characteristics. It was found that chemical modification of MWNTs in sulfuric and dichromic acids promotes the formation of functional groups on the material surface, inducing the enhancement of sensor response. The efficiency of acid treatment of MWNTs depends on the initial sample. For the MWNT-1020 sample (preliminary treated by the manufacturer), treatment in 6 M sulfuric acid was better (5.16–15.07%) in terms of their sensing behavior compared to the other type of MWNTs. This is due to the fact that during the functionalization of MWNT-1020 in dichromic acids, adsorption of dichromate ions on the surface of the material is observed, that blocks the adsorption of NO_2_. At the same time, for MWNT-1020 treated in sulfuric acid, according to EDX data, and the level of defectiveness is commensurate with the initial one, according to Raman spectroscopy. This makes it possible to conclude that, in this case, defects and functional group formation mechanisms prevail for NO_2_ adsorption. For MWNT-4060 nanotubes that were not chemically treated by the manufacturer, the treatment in 6 M dichromic acid was the best (3.95–13.27%) in terms of sensing properties. The fact that chemical treatment has different effects for different types of MWNTs indicates the importance of type of the material itself subjected to treatment, its morphology, and electronic properties. All these factors have a strong influence on the sensitivity of the room temperature NO_2_ gas sensor. This work shows that the compacted MWNTs can be successfully used as gas sensors as well as films.

## Figures and Tables

**Figure 1 micromachines-13-01495-f001:**
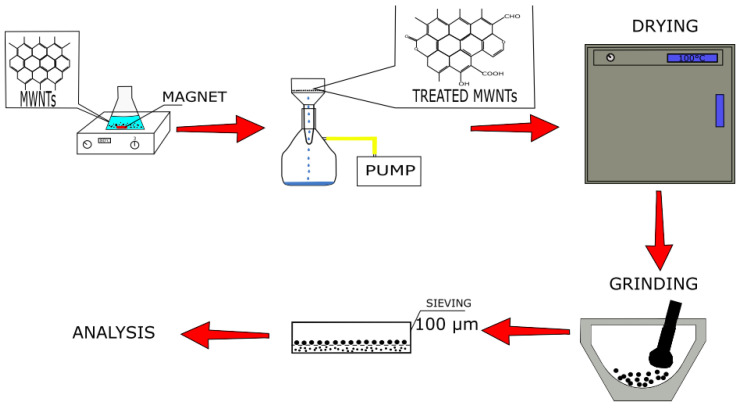
A schematic representation of research.

**Figure 2 micromachines-13-01495-f002:**
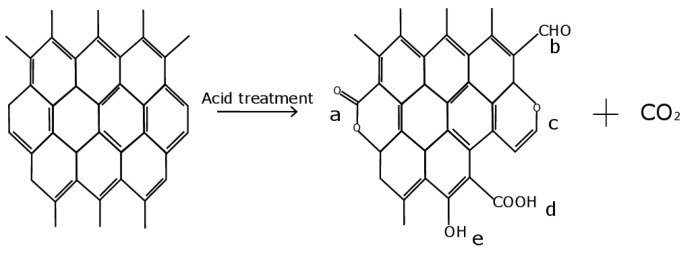
Functional groups formed by oxidation of MWNTs: (a) lactone; (b) carbonyl; (c) ester; (d) carboxyl; (e) hydroxyl (phenolic) groups.

**Figure 3 micromachines-13-01495-f003:**
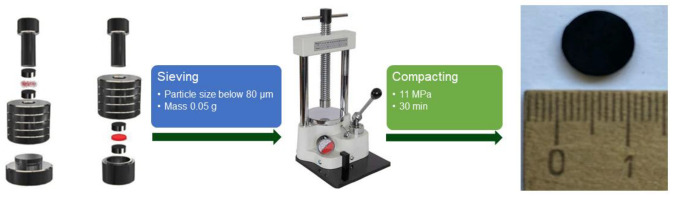
Scheme of preparation of pellets.

**Figure 4 micromachines-13-01495-f004:**
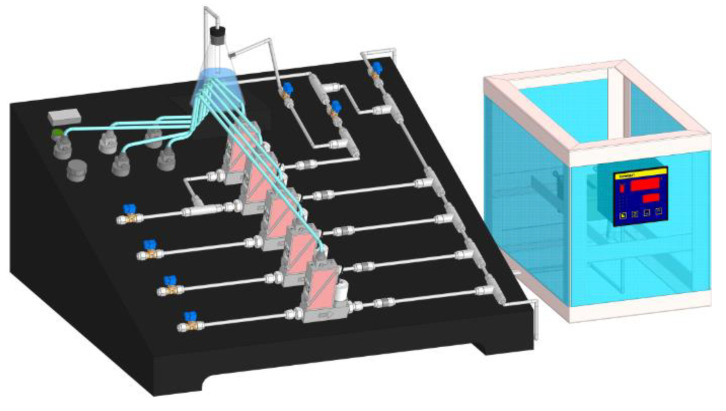
Scheme of a station for gas sensor testing.

**Figure 5 micromachines-13-01495-f005:**
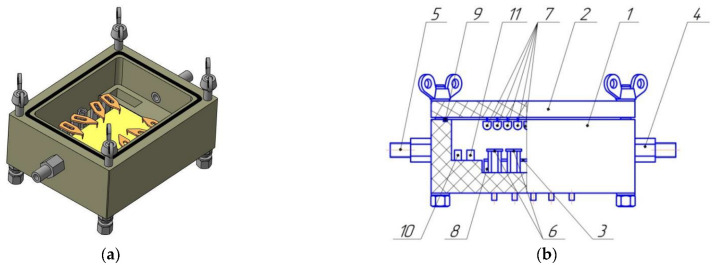
Model of measuring cell for testing of gas sensors (**a**) and its scheme (**b**): housing (1), cover (2), common contact (3), gas inlet (4), gas outlet (5), clamping contacts (6), LEDs (7), heater (8), seal (9), humidity-temperature sensor (10), pressure sensor (11).

**Figure 6 micromachines-13-01495-f006:**
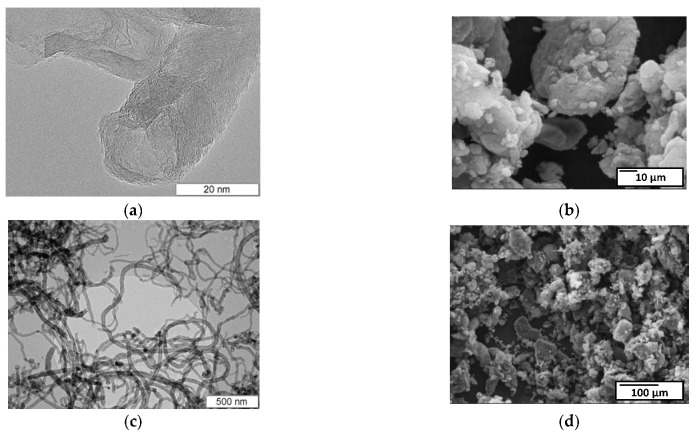
TEM (**a**,**c**) (Reprinted/adapted with permission from Ref. [51]) and SEM (**b**,**d**) micrographs of the MWNT-1020 sample.

**Figure 7 micromachines-13-01495-f007:**
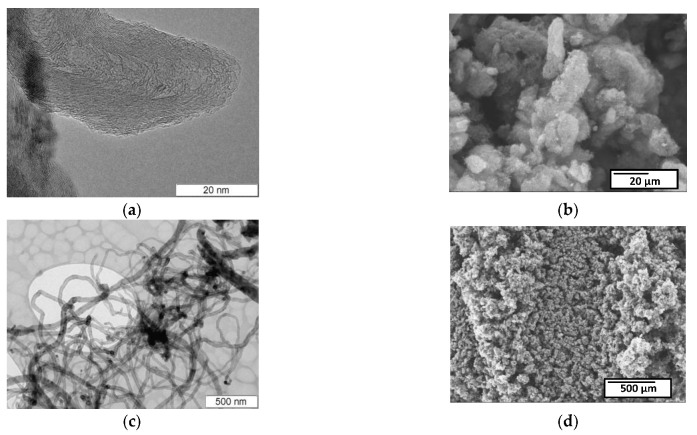
TEM (**a**,**c**) and SEM (**b**,**d**) micrographs of the MWNT-4060 sample.

**Figure 8 micromachines-13-01495-f008:**
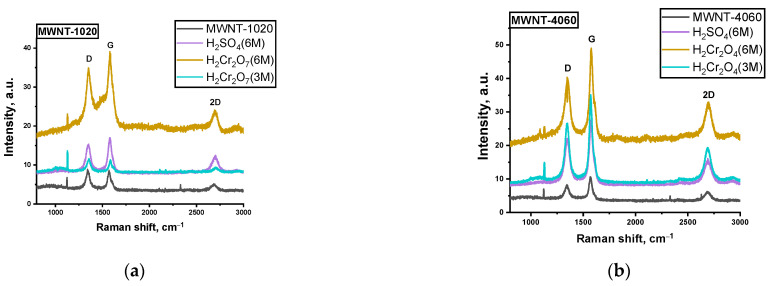
Raman spectra of initial and treated MWNT-1020 (**a**) and MWNT-4060 (**b**) samples.

**Figure 9 micromachines-13-01495-f009:**
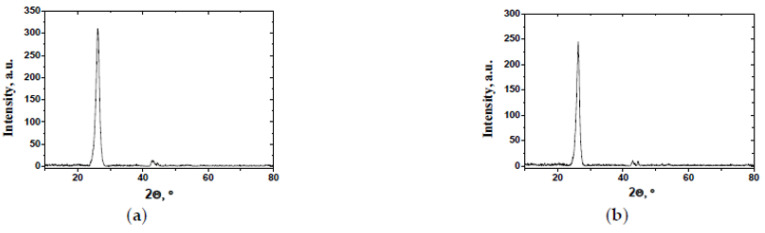
X-ray diffraction patterns of MWNT-1020 (**a**) and MWNT-4060 (**b**) samples.

**Figure 10 micromachines-13-01495-f010:**
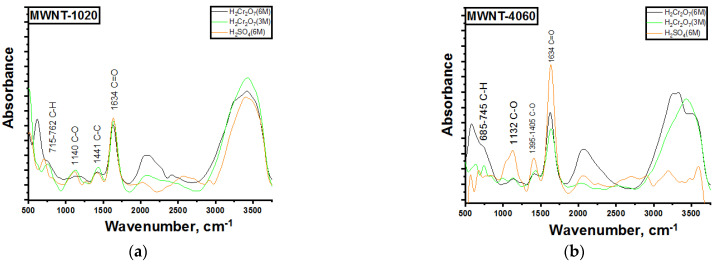
FTIR spectra of treated MWNT-1020 (**a**) and MWNT-4060 (**b**).

**Figure 11 micromachines-13-01495-f011:**
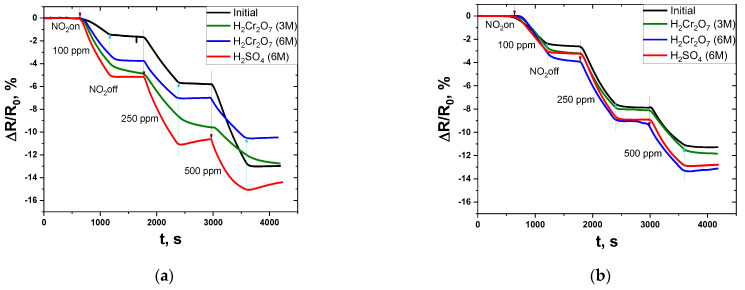
Response of pure and treated MWNT-1020 (**a**) and MWNT-4060 (**b**) to 100–500 ppm NO_2_ at room temperature (25 ± 2 °C).

**Table 1 micromachines-13-01495-t001:** I(D)/I(G) ratio (Raman spectroscopy) of the MWNTs.

Acid	Sample	I(D)/I(G)
-	MWNT-1020	1.04
	MWNT-4060	0.56
H_2_Cr_2_O_7_ (3 M)	MWNT-1020	0.85
	MWNT-4060	1.46
H_2_Cr_2_O_7_ (6 M)	MWNT-1020	0.78
	MWNT-4060	0.66
H_2_SO_4_ (6 M)	MWNT-1020	0.82
	MWNT-4060	0.74

**Table 2 micromachines-13-01495-t002:** Interlayer spacing and degrees of graphitization of initial MWNTs.

Acid	Sample	Interlayer Spacing d002, nm	Degree of Graphitization, %
-	MWNT-1020	0.34	46.5
-	MWNT-4060	0.3387	61.6
H_2_Cr_2_O_7_ (3 M)	MWNT-1020	0.33931	54.5
	MWNT-4060	0.34026	43.5
H_2_Cr_2_O_7_ (6 M)	MWNT-1020	0.34292	12.6
	MWNT-4060	0.33904	58.1
H_2_SO_4_ (6 M)	MWNT-1020	0.33967	50.3
	MWNT-4060	0.3386	62.8

**Table 3 micromachines-13-01495-t003:** Concentration of elements (in at.%) according to EDX.

Acid	Sample	EDX		XPS	
		C:O (at.)	Other Elements	C:O (at.)	Other Elements
-	MWNT-1020	Oxygen was not detected	Ni (0.16)	Oxygen was not detected	-
-	MWNT-4060	Oxygen was not detected	Ni (0.34)	Oxygen was not detected	-
H_2_Cr_2_O_7_ (3 M)	MWNT-1020	14	Cr (0.31)	n/a	n/a
	MWNT-4060	33	Ni (0.11), Cr (0.15)	n/a	n/a
H_2_Cr_2_O_7_ (6 M)	MWNT-1020	10	Cr (1.62)	8.55	Cr (0.18)
	MWNT-4060	17	Cr (0.48)	8.53	Cr (0.1)
H_2_SO_4_ (6 M)	MWNT-1020	82	S (0.07)	12.84	S (0.1)
	MWNT-4060	127	Ni (0.25), S (0.02)	8.78	S (1.15)

**Table 4 micromachines-13-01495-t004:** Response of sensors based on initial and treated MWNTs at room temperature.

Acid	Sample	ΔR/R_0_ (%)			R_25 °C_, Ω
		100 ppm	250 ppm	500 ppm	
-	MWNT-1020	1.7	5.8	13.0	1.39
	MWNT-4060	2.6	7.9	11.3	2.52
H_2_Cr_2_O_7_ (3 M)	MWNT-1020	4.9	9.6	12.8	1.76
	MWNT-4060	3.2	8.1	11.8	3.27
H_2_Cr_2_O_7_ (6 M)	MWNT-1020	3.8	7.1	10.5	5.14
	MWNT-4060	3.9	9.3	13.3	4.57
H_2_SO_4_ (6 M)	MWNT-1020	5.2	11.1	15.1	1.62
	MWNT-4060	3.3	8.9	12.9	1.42

**Table 5 micromachines-13-01495-t005:** Response time of sensors based on initial and treated MWNTs at room temperature.

Acid	Sample	Response Time (s)		
		100 ppm	250 ppm	500 ppm
-	MWNT-1020	902	535	511
	MWNT-4060	667	533	341
H_2_Cr_2_O_7_ (3 M)	MWNT-1020	687	619	487
	MWNT-4060	665	508	427
H_2_Cr_2_O_7_ (6 M)	MWNT-1020	590	437	374
	MWNT-4060	583	523	368
H_2_SO_4_ (6 M)	MWNT-1020	551	457	279
	MWNT-4060	590	514	347

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
