# Peer review of "NO2 Sensing Behavior of Compacted Chemically Treated Multi-Walled Carbon Nanotubes"

_micromachines, 2022, doi:10.3390/mi13091495_

Round 1

Reviewer 1 Report

The present study investigated the sensing behavior of chemically treated
Multi-walled carbon nanotubes (MWNTs) for the detection of NO2. The experimental part and results are sound. I have some major comments before the publication of this paper:

1- The quality of the figures in the manuscript is low. Especially in Figure 3, I suggest authors use Photoshop to edit the images.

2- In the introduction part, more recent articles should be cited to elucidate the potential of carbon nanomaterials:

10.1016/j.carbon.2022.03.040

10.1007/s12274-019-2438-0

3- The authors should measure the C:O ratio using XPS. This is the most confirmed method for such calculation.

4- What is the response time of the developed sensor?

5- How the LOD of the sensor was calculated?

6- The authors should study the effect of possible interferants in the sensing experiment to validate the practical applicability of the developed sensor.

Author Response

Reviewer 1.

The present study investigated the sensing behavior of chemically treated Multi-walled carbon nanotubes (MWNTs) for the detection of NO2. The experimental part and results are sound. I have some major comments before the publication of this paper:

1- The quality of the figures in the manuscript is low. Especially in Figure 3, I suggest authors use Photoshop to edit the images.

Thank you for the remark. The quality of the figures was improved.  Additional files with separate Figures were attached through submission.

2- In the introduction part, more recent articles should be cited to elucidate the potential of carbon nanomaterials:

10.1016/j.carbon.2022.03.040

10.1007/s12274-019-2438-0

These references were added in the Introduction.

3- The authors should measure the C:O ratio using XPS. This is the most confirmed method for such calculation.

We added XPS data for 6 from the 8 samples. Unfortunately, the equipment malfunctioned before we had time to obtain data for the samples treated in H2Cr2O7 3M.

4- What is the response time of the developed sensor?

Thank you for the question! If the response time is the time needed to achieve 90 % of sensor response value from its initial value at the certain concentration, the response time was estimated, the values are shown in the table

Acid

Sample

Response time (s)

100 ppm

250 ppm

500 ppm

-

MWNT-1020

902.487

535.4

511.168

MWNT-4060

666.907

533.241

340.856

H2Cr2O7 (3M)

MWNT-1020

687.098

618.665

486.158

MWNT-4060

664.804

507.594

427.287

H2Cr2O7 (6M)

MWNT-1020

590.408

437.161

373.694

MWNT-4060

582.944

522.636

368.138

H2SO4 (6M)

MWNT-1020

550.816

457.096

278.855

MWNT-4060

589.561

514.465

347.213

5- How the LOD of the sensor was calculated?

The LOD was not determined in this paper.

The main task of this work was to estimate the effect of chemical modification of MWNTs on their sensing properties. Therefore for the measurements we developed a technique, including three cycles: 100 ppm, 250 ppm and 500 ppm. Of course, these sensors can detect a lower concentration of analytes. It is related to the high response of sensor toward 100 ppm of analyte’s concentration. However the LOD of these sensors wasn’t determined.

6- The authors should study the effect of possible interferants in the sensing experiment to validate the practical applicability of the developed sensor.

Thank you for the comment! It is true, that for the cross-selectivity evaluation it is necessary to evaluate the properties of the sensors with respect to other gases. These sensors were tested on ammonia. The choice of ammonia is due to the donor interaction with MWNTs. The cross-sensitivity (selectivity estimation) of sensors was examined (Supplementary materials). These sensors were tested on detection of model gases (ammonia and methane) at room temperature. The choice of ammonia was based on electron donating interaction with MWNTs (that is different compared to NO2 sensing mechanism). From the graphs one can see that these sensors have a low response to 100-500 ppm of ammonia compared to the 100-500 ppm of nitrogen dioxide (Supplementary materials, Fig. S1). It is also worth noting that the plots of change in sensor resistance at the time of ammonia feeding have a positive slope, while the plots of change in sensor resistance at the time of nitrogen dioxide showing the negative slope. This is explained by the fact that adsorption of ammonia on the surface of the MWNTs reduces the number of charge carriers, which contributes to an increase in the resistance of the material. Chemical modification of the initial MWNT samples led to a change in the sensing properties. Chemically modified MWNT-1020 showed a greater response compared to the non-treated one. Treatment in 6 M sulfuric acid and 3 M dichromic acid increased the response approximately by a factor of 3. This is due to the fact that treatment in these acids increases the number of functional groups, which is reflected in an increase in the degree of graphitization and an increase in the C to O ratio (according to EDX and XPS). Chemical modification of MWNT-4060 in sulfuric and dichromic acids (6M) also led to an improvement in the sensor response. A comparison of methane detection by compacts showed that there was no significant response to CH4 at room temperature at 100-500 ppm concentration (Supplementary materials, Fig. S2). The absence of response was also detected for chemically treated samples.

The testing of other gases was not carried out since our installation was not designed to use any other gases excluding NO2, NH3, CH4, air.

Reviewer 2 Report

This work reported the sensing behavior of chemically treated multi-walled carbon nanotubes (MWNTs) at room temperature. The structure and performance were investigated thoroughly. The discussion and analysis of measurement results are sufficient for supporting the understanding of mechanism and innovation. The reviewer thinks that it can be considered for potential publication after minor revision.

1. The sensing detection methods have played important role in the field of rapid, sensitive and selective analysis. The introduction part can be further improved by citing recent research works. The following papers can be involved: 

https://doi.org/10.1016/j.micromeso.2020.110583

https://doi.org/10.1016/j.foodchem.2020.128484

https://doi.org/10.1016/j.jelechem.2018.02.028

https://doi.org/10.1016/j.msec.2021.111982

https://doi.org/10.1016/j.microc.2020.105234

https://doi.org/10.1016/j.clay.2020.105907

2. The purity of reagents and materials?

3. Figure 6 and Figure 7, the SEM images at high magnification and TEM images at low magnification are important for the clearly observation.

4. The authors can provide the repeatability and long-term stability.

5. It is better to provide the effect of other gases on the NO2 sensing behavior. 

Author Response

Reviewer 2.

This work reported the sensing behavior of chemically treated multi-walled carbon nanotubes (MWNTs) at room temperature. The structure and performance were investigated thoroughly. The discussion and analysis of measurement results are sufficient for supporting the understanding of mechanism and innovation. The reviewer thinks that it can be considered for potential publication after minor revision.

  1. The sensing detection methods have played important role in the field of rapid, sensitive and selective analysis. The introduction part can be further improved by citing recent research works. The following papers can be involved: 

https://doi.org/10.1016/j.micromeso.2020.110583

https://doi.org/10.1016/j.foodchem.2020.128484

https://doi.org/10.1016/j.jelechem.2018.02.028

https://doi.org/10.1016/j.msec.2021.111982

https://doi.org/10.1016/j.microc.2020.105234

https://doi.org/10.1016/j.clay.2020.105907

These references were added in the introduction.

  1. The purity of reagents and materials?

The information was added. “A 6M sulfuric acid solution was prepared from concentrated sulfuric acid of 94.6% purity. H2Cr2O7 solutions were prepared from solid chemically pure chromium oxide (VI), 99.0% purity.”

  1. Figure 6 and Figure 7, the SEM images at high magnification and TEM images at low magnification are important for the clearly observation.

Thank you for the advice! The TEM and SEM images at different magnification were added.

  1. The authors can provide the repeatability and long-term stability.

Thank you for the comment! The data on repeatability and long-term stability of sensors were not investigated in this paper, since the accent was paid to chemical treatment effect and the possibility to use the pellets of MWNTs for NO2 detection. These characteristics will be presented in the our next paper devoted to analysis of kinetics and noise analysis of pellets. 

  1. It is better to provide the effect of other gases on the NO2 sensing behavior. 

Thank you for the comment! It is true, that for the selectivity’s evaluation it is necessary to evaluate the properties of the sensors with respect to other gases. These sensors were tested on ammonia. The choice of ammonia is due to the donor interaction with MWNTs. The cross-sensitivity (selectivity estimation) of sensors was examined (Supplementary materials). These sensors were tested on detection of model gases (ammonia and methane) at room temperature. The choice of ammonia was based on electron donating interaction with MWNTs (that is different compared to NO2 sensing mechanism). From the graphs one can see that these sensors have a low response to 100-500 ppm of ammonia compared to the 100-500 ppm of nitrogen dioxide (Supplementary materials, Fig. S1). It is also worth noting that the plots of change in sensor resistance at the time of ammonia feeding have a positive slope, while the plots of change in sensor resistance at the time of nitrogen dioxide showing the negative slope. This is explained by the fact that adsorption of ammonia on the surface of the MWNTs reduces the number of charge carriers, which contributes to an increase in the resistance of the material. Chemical modification of the initial MWNT samples led to a change in the sensing properties. Chemically modified MWNT-1020 showed a greater response compared to the non-treated one. Treatment in 6 M sulfuric acid and 3 M dichromic acid increased the response approximately by a factor of 3. This is due to the fact that treatment in these acids increases the number of functional groups, which is reflected in an increase in the degree of graphitization and an increase in the C to O ratio (according to EDX and XPS). Chemical modification of MWNT-4060 in sulfuric and dichromic acids (6M) also led to an improvement in the sensor response. A comparison of methane detection by compacts showed that there was no significant response to CH4 at room temperature at 100-500 ppm concentration (Supplementary materials, Fig. S2). The absence of response was also detected for chemically treated samples.

Round 2

Reviewer 1 Report

The authors have implemented the required revision, and I believe the paper is ready to publish in Micromachines.